# Lossy Mode Resonance Based Microfluidic Platform Developed on Planar Waveguide for Biosensing Applications

**DOI:** 10.3390/bios12060403

**Published:** 2022-06-10

**Authors:** Melanys Benítez, Pablo Zubiate, Ignacio Del Villar, Abián B. Socorro-Leránoz, Ignacio R. Matías

**Affiliations:** 1Department of Electrical, Electronic and Communication Engineering, Public University of Navarra, Ed. Los Tejos, Campus Arrosadía s/n, E-31006 Pamplona, Spain; melanys.benitez@unavarra.es (M.B.); pablo.zubiate@unavarra.es (P.Z.); ignacio.delvillar@unavarra.es (I.D.V.); ab.socorro@unavarra.es (A.B.S.-L.); 2Institute of Smart Cities, Public University of Navarra, Ed. Jerónimo de Ayanz, Campus of Arrosadía s/n, E-31006 Pamplona, Spain; 3Navarra Institute for Health Research (IdiSNa), Recinto de Complejo Hospitalario de Navarra, C/Irunlarrea, 3, E-31008 Pamplona, Spain

**Keywords:** flow cell, lossy mode resonance, microfluidics, biosensing platform, planar waveguide, biosensors

## Abstract

The development of resonance phenomena-based optical biosensors has gained relevance in recent years due to the excellent optical fiber properties and progress in the research on materials and techniques that allow resonance generation. However, for lossy mode resonance (LMR)-based sensors, the optical fiber presents disadvantages, such as the need for splicing the sensor head and the complex polarization control. To avoid these issues, planar waveguides such as coverslips are easier to handle, cost-effective, and more robust structures. In this work, a microfluidic LMR-based planar waveguide platform was proposed, and its use for biosensing applications was evaluated by detecting anti-immunoglobulin G (anti-IgG). In order to generate the wavelength resonance, the sensor surface was coated with a titanium dioxide (TiO_2_) thin-film. IgG antibodies were immobilized by covalent binding, and the detection assay was carried out by injecting anti-IgG in PBS buffer solutions from 5 to 20 μg/mL. The LMR wavelength shifted to higher values when increasing the analyte concentration, which means that the proposed system was able to detect the IgG/anti-IgG binding. The calibration curve was built from the experimental data obtained in three repetitions of the assay. In this way, a prototype of an LMR-based biosensing microfluidic platform developed on planar substrates was obtained for the first time.

## 1. Introduction

Progress in the development of biosensors has been a strong support in many areas such as medical diagnosis, food safety, and environmental monitoring [1]. The development of resonance-based optical fiber sensors for detecting biomolecules has gained relevance lately due to its interesting properties, such as small size, immunity to electromagnetic interference, and progress in the research on different materials that can be used for fabricating sensors [2].

By depositing a thin film of certain materials on either fiber optics or standard planar waveguides, such as coverslips [3] or glass slides [4], it is possible to generate resonant phenomena known as lossy mode resonances (LMRs). This technology has emerged as a very interesting and promising sensing platform [5,6]. As it is well-known, LMRs occur when the real part of the thin film permittivity is positive and higher in magnitude than both its own imaginary part and the material surrounding the thin film. According to this, materials that can induce this phenomenon are typically metal oxides and polymer coatings, which are less expensive than the metallic materials typically used in surface plasmon resonance (SPR)-based sensors. In addition, LMR enables one to tune the position of the resonance in the optical spectrum, to excite the resonance with both transverse electric (TE) and transverse magnetic (TM) polarized light, and to generate multiple resonances [7].

Among other oxides, titanium dioxide (TiO_2_) is often used as a high refractive index coating for optical sensors [8]. Besides, it offers other interesting physical properties such as high hardness, low optical absorption, and biocompatibility [9]. TiO_2_ has already been used in photodynamic therapy for cancer, drug delivery systems, cell imaging, biosensors for biological assays, and genetic engineering [10,11].

Despite the advantages of using optical fibers as substrates in LMR-based sensors, they require the splicing of the sensor head for transmission configuration. Moreover, because LMRs can be obtained in both TE and TM polarization, it is a challenging aspect in this technology to decrease, as much as possible, the resonance bandwidth. Otherwise, resonances corresponding to TE and TM modes could appear overlapped or wider, which reduces the resolution of the measurements [3]. The most common strategy that is followed to solve this is to polarize the system. So far, the best LMR-based biosensing platforms use a D-shaped fiber, which, thanks to its asymmetric cross-section, separates both modes with the aid of an in-line polarizer and a polarization controller or by using a polarization-maintaining fiber [12]. However, D-shaped optical fibers are expensive, and controlling the polarization is difficult.

As an alternative to avoid these issues, planar waveguides, such as glass coverslips, are easier-to-handle, cost-effective, and more robust structures. LMRs can be generated on this type of substrate and easily polarized by using a linear polarizer. It has been demonstrated that using coverslips, instead of other glass planar substrates such as slides, allows the generation of LMRs with better characteristics such as resonance depth, a more directive coupling, and even the possibility of multiparameter detection [13,14].

On the other hand, the discipline of microfluidics has become more popular during the last years due to the micro-miniaturized analytical equipment for biological analysis, chemical sensing, genetic analysis, and metabolic monitoring and detection. The advantages of such miniaturized devices include small reagent volumes, fast response times, low costs, and the reduction or elimination of cross-contamination [15]. In this sense, planar technology, such as the one proposed in this work, could be more suitable for point-of-care applications, where specialists are more familiar with one-use devices that can be easily replaced [16,17]. The integration of microfluidics with biosensor technology offers new opportunities for future applications with improvements in portability, real-time detection, higher accuracy, increased sensitivity, and simultaneous analysis of different analytes in a single device. In the field of optical biosensors, we find a variety of devices integrated into microfluidic systems, some of them based on long-period grating (LPG) [18], Mach-Zehnder interferometry [19], or SPR [20,21,22].

In this work, a novel prototype of an LMR-based biosensing microfluidic platform developed on planar substrates is presented. Some considerations regarding the light propagation in the proposed substrate were analyzed, together with a general assessment of the whole system performance. Finally, some conclusions were extracted regarding the benefits and possible improvements of the system, as well as on the use of these substrates as LMR-based biosensors.

## 2. Experimental Section

### 2.1. Chemical Reagents

Methacrylic acid/methacrylate copolymer (Eudragit L100) was purchased from Evonik Health Care (Essen, Germany. 1-ethyl-3-[3-dimethylaminopropyl] carbodiimide hydrochloride (EDC) and N-hydroxysuccinimide (NHS) were purchased from Thermo Fisher Scientific (Waltham, MA, USA). Ethanol (EtOH), phosphate-buffered-saline (PBS, 10 mM, pH 7.4), D-(+)-Glucose, bovine serum albumin (BSA), rabbit IgG, and anti-Rabbit IgG were purchased from Sigma Aldrich Inc. (St. Louis, MO, USA).

### 2.2. Sensor Fabrication and Experimental Setup

A soda-lime glass coverslip (18 mm × 18 mm × 0.15 mm) was coated with a thin film of TiO_2_ (70 nm thickness) by the Atomic Layer Deposition (ALD) technique, using the Savannah G2 ALD System from Veeco Inc. (Plainview, NY, USA). This deposition method generates thin films with atomic scale precision and it is based on the gas-solid reactions occurring at the surface of the substrate. The majority of ALD reactions use two chemicals, typically called precursors [23]. In this case, for the TiO_2_ deposition processes, water and Tetrakis(dimethylamido)titanium (TDMAT) were used as oxygen and titanium precursors, respectively. Each cycle included four steps: a 0.1-s pulse of TDMAT, a purge of the reaction chamber for 10 s to remove the non-reacted precursor and reaction by-products, a 0.015-s pulse of water, and again a 10-s purge step. Temperature during the process was set to 100 °C. This coating allowed the generation of the LMR in the visible range of the optical spectrum.

A picture of the experimental setup is shown in Figure 1. Light from an ASBN-W tungsten-halogen broadband source from Spectral Products Inc. (Putnam, FL, USA) was launched into a multimode optical fiber from Ocean Optics (200/225 µm of core/cladding diameter). This fiber was placed in front of one of the lateral sides of the coverslip, which acted as a planar waveguide. The output light of the waveguide passed through a polarizer and was then received by another multimode fiber, which was connected to an HR4000 spectrometer (OceanInsight^®^), operating in the visible-NIR wavelength range, between 200 and 1100 nm.

A temperature control system acquired the temperature from a thermistor and drove the current to two Peltier cells in order to adjust the temperature at 26 °C (±0.05 °C) [24]. The microfluidic system was connected to a peristaltic pump that allowed to pump the appropriate solution into the flow cell.

### 2.3. Analysis of the Optical Field Intensity Distribution

The thickness of the TiO_2_ thin film was based on the analysis of the propagation through the coverslip waveguide obtained with FIMMPROP, an integrated module of FIMMWAVE (Photon Design Inc., Oxford, UK). The finite difference method (FDM) with the Quasi 2D version, was used to calculate the modes and the fields in the cross section of the waveguide for a total number of 30 modes, which provided good convergence in the results. In addition, a Gaussian source of 200 µm full width at half maximum was used according to the 200 µm multimode fiber used in the experiments for exciting the planar waveguide. For a TiO_2_ thin film of 70 nm, the first LMR at TM polarization was generated in air (n = 1.0002) [25] at 560 nm, whereas in water (n = 1.333) [12], the LMR was centered at 776 nm (see Figure 2). This was considered the optimum design because, according to LMR theory, the first LMR, either at TE or TM polarization, is the most sensitive [22], and the sensitivity increases for the same LMR as a function of wavelength [21]. The wavelength range of the spectrometer was from 400 to 1000 nm. As it can be observed, the LMR was located at a long wavelength but was still far from the spectrometer upper limit. This avoids running the risk of not being able to monitor the LMR during the bioassay.

Regarding the refractive index of the waveguide, since both the microscope slides and the coverslips were made of soda lime glass, the refractive index model of [26,27,28] was used. The planar waveguide was placed on a poly(methyl methacrylate) (PMMA) substrate material, whose refractive index was modeled according to [27]. In addition, the refractive index of TiO_2_ ranges from 2.42 to 2.16 in the wavelength range 400–1000 nm, according to the results obtained in the ellipsometer UVISEL 2 from Horiba, with a spectral range of 0.6–6.5 eV (190–2100 nm), an angle of incidence of 70°, a spot size of 1 mm, and software Delta Psi 2 version 2.9 (from HORIBA France SAS, Palaiseau, France).

Finally, in Figure 3 theoretical analysis of the optical field intensity distribution for the first four coverslip TM modes, TM_0_, TM_1_, TM_2_, and TM_3_, is presented. TM_3_ was transformed into the shape of TM_2_, TM_2_ into TM_1_, and TM_1_ into TM_0_. All this occurred because TM_0_ experienced a transition to guidance in the thin film as the wavelength decreased from 760 to 730 nm, according to the position of the LMR in Figure 2.

### 2.4. Flow Cell

The PMMA microfluidic cell is described in Figure 4. It was composed of two parts (bottom and top) whose dimensions were 100 mm × 35 mm × 4 mm. The top part contained the holes for inlet and outlet fluids and a flow channel of 10 mm × 1 mm × 1 mm, through which a liquid volume of 10 μL circulated. The bottom part contained two narrow slots for placing the optical fibers and a rectangular slot, in which the sensor was placed so that the fibers and the coverslip were aligned. It also contained a hole to insert a thermistor in order to control the temperature during the experiments. Both the top and bottom parts contained a hole across the structure where the polarizer was placed. In this way, when the cell parts are put together, the flow channel fits perfectly over the sensor slot.

### 2.5. Application of the Microfluidic System for Anti-IgG Detection

The LMR-based microfluidic system using a TiO_2_ thin-film on a planar waveguided was used to detect immunoglobulin G (IgG)/anti-IgG interactions as an example to demonstrate its potential. IgGs are cost-effective and useful biomolecules used to evaluate the performance of biosensors [29]. The operation principle consisted of monitoring, in real time, the interaction between an antigen (in this case, an anti-IgG) and its specific antibody (IgG) grafted on the coverslip surface, which induces changes in the thickness of the coating and in the effective RI of the sensing layer. An accurate determination of this change can be attained by tracking the wavelength shift of the LMR.

#### Surface Functionalization and Assay Protocol

Once the coverslip was coated with the metal oxide, which allowed the LMR generation, it was immersed in 2 mM (0.04% *w*/*v*) Eudragit L100 copolymer in ethanol for 1 min and then it was left drying in air for at least 15 min until the solvent was completely evaporated. The carboxylic functional groups, necessary for the IgG covalent binding on the sensor surface, were provided by the Eudragit. When the sensor was placed inside the microfluidic system, an EDC/NHS solution (2 mM/5 mM, respectively) was injected and flowed over the Eudragit layer at 14 µL/min [30,31].

The covalent IgG immobilization on the activated surface was carried out immediately, by injecting a solution of 1 mg/mL IgG in PBS for an hour at a flow rate of 7 µL/min. Unbound IgGs were then removed by washing with PBS buffer for 10 min at a flow rate of 20 μL/min. For passivating the surface, BSA 1% (*w*/*v*) was injected for 15 min at 14 µL/min to block unreacted remaining active carboxylic groups and to prevent nonspecific adsorption onto the sensing surface.

Once the antibody was immobilized on the coverslip, the assay was completed by injecting different concentrations of anti-IgG (5, 10, 15 and 20 μg/mL). Each anti-IgG concentration was injected for 15 min at a flow rate of 7 µL/min, and PBS was circulated for 5 min at 20 μL/min between each anti-IgG concentration to measure and to evaluate the effect of the bioreceptor-analyte binding interaction on the surface. A schematic representation of the surface biofunctionalization and analyte detection is shown in Figure 5.

## 3. Results and Discussion

### 3.1. LMR Obtained with TiO_2_ Thin-Film

With the deposition of the 70-nm TiO_2_ thin-film on the coverslip surface by ALD technique, it was possible to generate the resonant phenomenon that constituted the optical detection principle of the proposed system. Figure 6 shows the representation of the LMR spectrum in air and water, in TM polarization. As observed, the resonance peak corresponding to air (n = 1) appeared in 586 nm and shifted 180 nm towards higher wavelengths when the sensor was exposed to water, a medium with higher refractive index (n = 1.33). This result agrees well with the analytical optical spectrum obtained with the simulations performed in Section 2.3.

### 3.2. System Stability at a Fixed Temperature

Considering that abrupt temperature changes affect the functionality of antibodies and, on the other hand, temperature variations produce LMR wavelength shifts, it is essential to keep this parameter constant during experiments. The system behavior was evaluated by setting the temperature at 26 °C for one hour while ultrapure water was flowing. The optical spectrum was registered, as well as the temperature values measured simultaneously by the thermistor. As shown in Figure 7, the temperature measured in the sensitive area fluctuated around the set value, with slight variations of ±0.05 °C. However, in spite of the fluctuations experienced by the temperature control system, the LMR central wavelength can be considered stable, since at the end of the test the total drift was lower than 0.5 nm. In this way, it can be considered that the proposed system is capable of remaining stable at a fixed set temperature throughout the detection test, in spite of small temperature fluctuations.

### 3.3. Flow Cell and Sensor Response to Refractive Index Variations

In order to evaluate the sensitivity of the system to the refractive index, glucose solutions at different concentrations (10%, 20%, 30% and 40% *w*/*v*) in ultrapure water were injected and flowed through the sensor. During this process, the LMR wavelength was monitored. The sensor was exposed to each glucose solution for 15 min, first increasing the concentrations and then decreasing them. Ultrapure water was circulated at the beginning and the end of the experiment for taking it as the baseline. Table 1 relates each glucose concentration to its corresponding RI value and to the wavelength shift that occurred in each case.

The sensor response is shown in Figure 8. It is observed in Figure 8a that the LMR wavelength shifted to higher values while increasing the refractive index. Moreover, it is evident that the microfluidic system was able to completely extract the liquid from the sensitive surface, without leaving any residue that could interfere with the detection of the current solution. Figure 8b shows the calibration curve, built from the sensor response to different RI corresponding to the glucose solutions mentioned above. The sensor was able to detect RI changes with a sensitivity of 1762.8 nm/RIU, and the data were adjusted to a linear response with R^2^ = 0.9982.

### 3.4. Anti-IgG Detection

This LMR-based planar waveguide microfluidic sensor system was used for detecting anti-IgG by carrying out a “label-free” assay when IgG specific antibodies were used as bioreceptors. In this sense, the spectrum during antibody immobilization on the sensor surface was monitored, as well as its behavior when passing the BSA solution to fill non-specific sites (see Figure 9). As it can be observed, the LMR wavelength increased from the very first instant the IgGs solution reached the sensor. During the first minutes of interaction, the curve showed an abrupt growth due to the complete availability of functional groups on the surface for IgG to bind and also due to the increase in the refractive index of the solution. Then, the slope gradually decreased, reflecting that the IgGs were binding progressively to the surface. BSA did not seem to have bound to the surface, since there was no change in the resonance wavelength, which indicates that the entire sensitive surface was covered by the IgGs. Once PBS was circulated through the system, the curve completely stabilized.

As in the immobilization step, the LMR wavelength was monitored in real time during detection to obtain a sensorgram (Figure 10). The sensorgram represents the change in the spectral position of the LMR over time, with respect to the baseline (PBS), when anti-IgG solutions are circulated in increasing concentrations, ranging from 5 to 20 μg/mL. According to the LMR theory, the increase of thickness in the thin film leads to a wavelength shift of the LMR to longer wavelengths [7]. This is what occurred when anti-IgGs adhered to the surface of the sensor, increasing the thickness of the structure deposited on the coverslip.

As expected, the antibody–antigen interaction induced an increase of the LMR wavelength and this effect can be explained considering an increase in both the thickness and the average RI of the deposited biolayer [32]. As can be seen, the resonance wavelength shifted to higher values as the anti-IgG concentration increased. A progressive displacement was observed as a function of time, meaning that the antigens were binding to the specific sites on the antibody. The change in the resonance wavelength that remained after washing with PBS was only related to the amount of anti-IgG that were captured by the bioreceptor on the coverslip surface, and they can be directly associated to the increasing concentration of anti-IgGs.

The calibration curve of the LMR-based planar waveguided biosensor is shown in Figure 11. It was obtained from three assays carried out under the same experimental conditions with three identical and independent sensors (n = 3).

The curve represents the wavelength shift of the LMR with respect to the baseline, depending on the concentration of anti-IgG. The experimental data were fitted by the Hill function, which is a well-accepted mathematical model that provides a way in which to quantify the degree of interaction between ligand binding sites [33]. As observed, the minimum shift of the resonant wavelength was 0.842 nm at 5 μg/mL, whereas the total shift was 7.446 nm.

According to the response curve obtained from the mean of the shifts for each analyte concentration in the three repetitions of the experiment, the sensor response fit the Hill equation with a correlation coefficient R^2^ of 0.9998. From the calibration curve, it is possible to extrapolate the limit of detection (LOD) of the biosensor, defined as the signal of the blank plus three times the standard deviation of the blank [34]. The LOD obtained was 2.2 μg/mL, considering σ_blank_ = 0.28 nm. Although this LOD value is considered high compared to those reached in other contributions where anti-IgG detection was performed, it is necessary to emphasize that the design of the experiments was not directed towards the optimization of this parameter, but rather to verify the functionality of the proposed system for the detection of biomolecules. The response time of the biosensor was 12.73 min, which can be considered comparable with other LMR-based biosensing platforms [34]. Regarding sensitivity, the value calculated from the calibration curve was 0.43 nm/(μg/mL).

Taking into account that planar substrate technologies are currently the most commercially attractive platforms for the mass production of biosensor devices, the system proposed in this work could be improved for use in the development of fast screening devices.

## 4. Conclusions

An LMR-based planar waveguided microfluidic platform was implemented in this work, and its potential for biosensing applications was demonstrated for the first time. The LMR was generated by depositing a TiO_2_ thin film onto a coverslip, according to simulations results. This approach proposed a robust and easy-to-handle structure, based not on intensity but on wavelength shifts, that solves the above-mentioned disadvantages of using optical fibers in LMR-based sensors. The stability of the system response at a set temperature, as well as the performance of the microfluidic cell when circulating different RI solutions, were evaluated. The coverslip surface was functionalized by covalent binding, employing Eudragit L100 copolymer. This method has the advantage of being time saving and forming strong and stable-over-time bonds.

Regarding biosensing, anti-IgG detection was carried out using the proposed system, since IgG is a widely used molecule for evaluating sensors performance. It was demonstrated that binding interactions between IgG and anti-IgG caused the LMR central wavelength to shift. Therefore, different anti-IgG concentrations in buffer solution in the range of 5 to 20 µg/mL were detected with an LOD of 2.2 µg/mL and a sensitivity of 0.43 nm/(µg/mL). Future works will focus on improving sensor performance and using it for detecting other biomolecules of interest with the same microfluidic biosensing platform.

## Figures and Tables

**Figure 1 biosensors-12-00403-f001:**
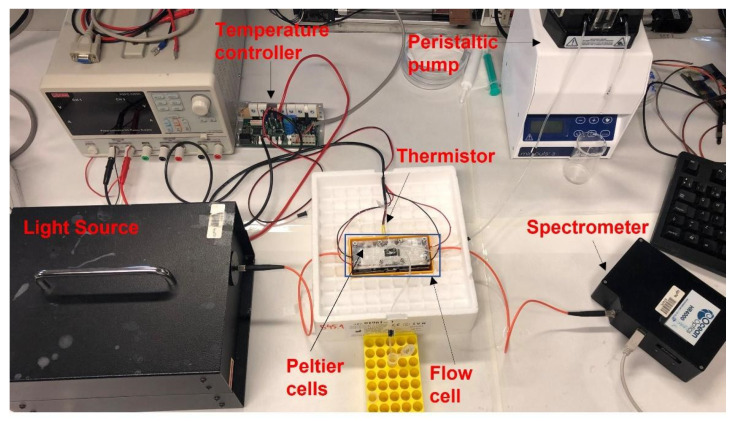
Picture of the experimental setup.

**Figure 2 biosensors-12-00403-f002:**
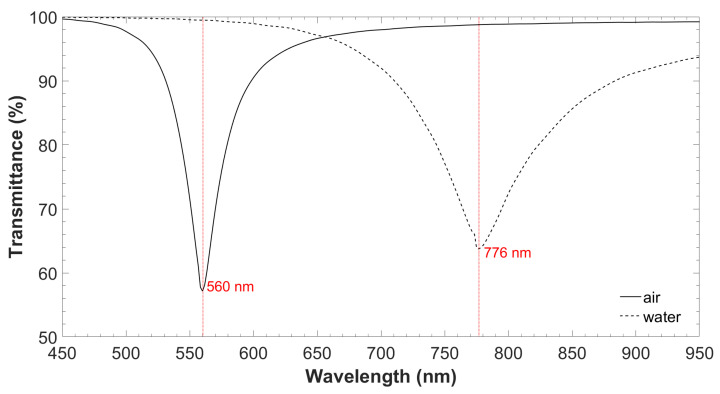
Theoretical optical spectrum in the air and water of the generated LMR by the ALD technique with TiO_2_ thin film on the coverslip.

**Figure 3 biosensors-12-00403-f003:**
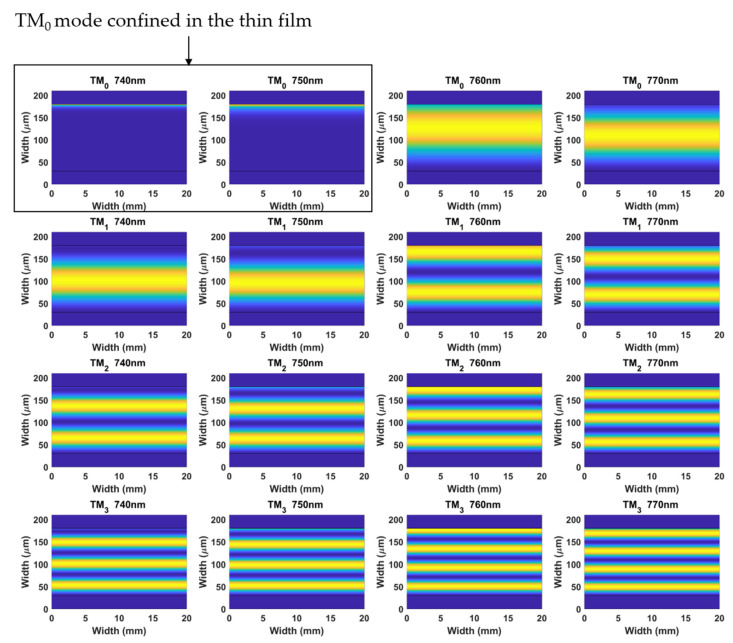
Optical field intensity distribution at the different wavelengths of TM_0_, TM_1_, TM_2_, and TM_3_, in the cross-section of the coverslip waveguide coated with a 70 nm TiO_2_ thin film. Surrounding refractive index: 1.333 (water).

**Figure 4 biosensors-12-00403-f004:**
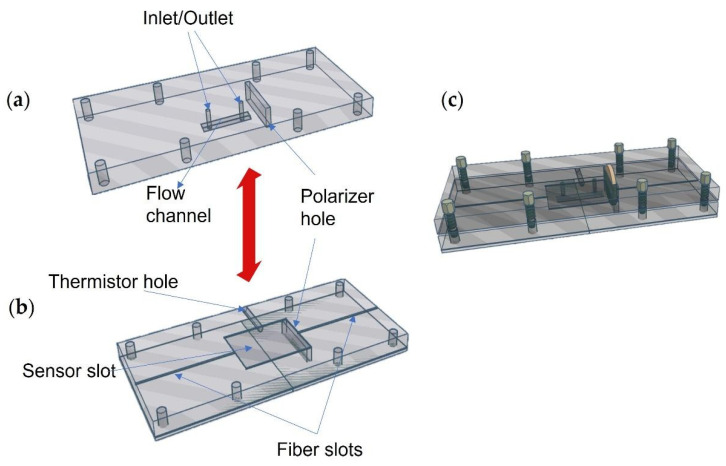
Schematic representation of the PMMA flow cell parts: (**a**) flow cell top, (**b**) flow cell bottom, and (**c**) screwed top and bottom parts with polarizer inside.

**Figure 5 biosensors-12-00403-f005:**
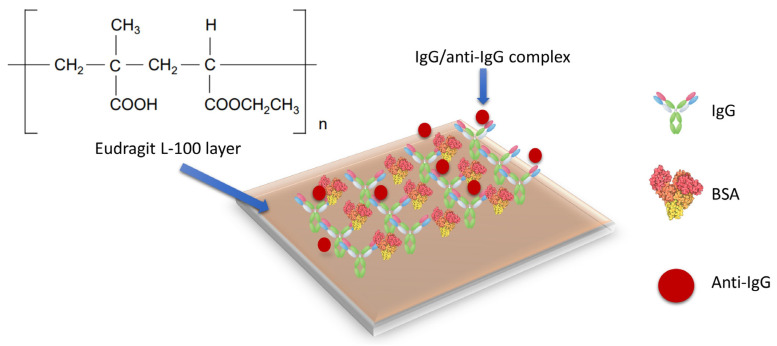
Sketch of the sensor functionalization and analyte detection.

**Figure 6 biosensors-12-00403-f006:**
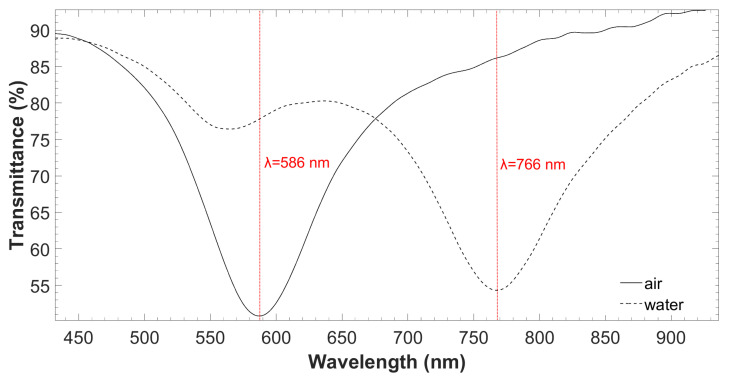
Experimental optical spectrum in the air and water of the generated LMR by the ALD technique with a TiO_2_ thin film on the coverslip.

**Figure 7 biosensors-12-00403-f007:**
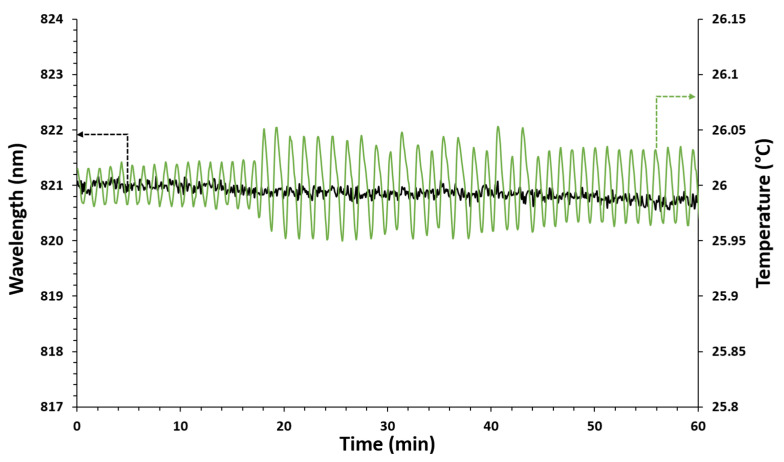
Stability of the system at a fixed temperature. Green line: temperature values measured by the thermistor inside the cell. Black line: LMR wavelength during the test.

**Figure 8 biosensors-12-00403-f008:**
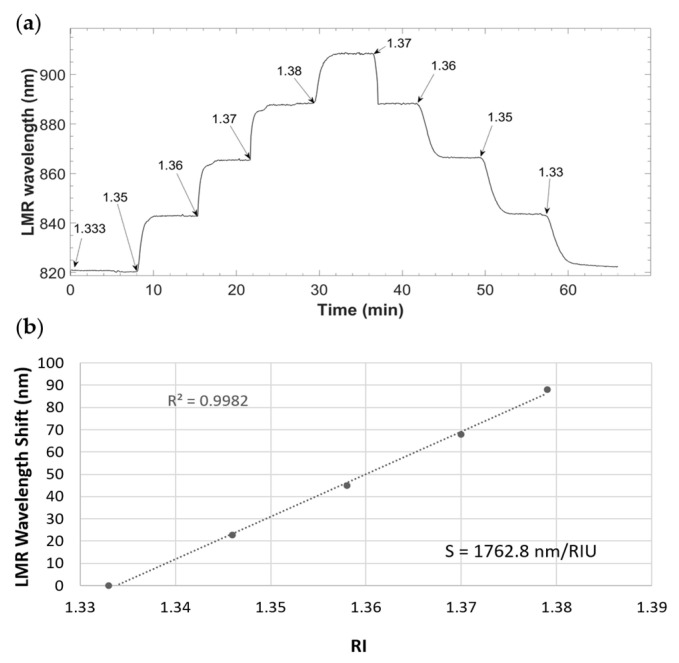
System response to the circulation of different refractive index glucose solutions: (**a**) sensorgram of different RI glucose solution detection, and (**b**) calibration curve of the sensor.

**Figure 9 biosensors-12-00403-f009:**
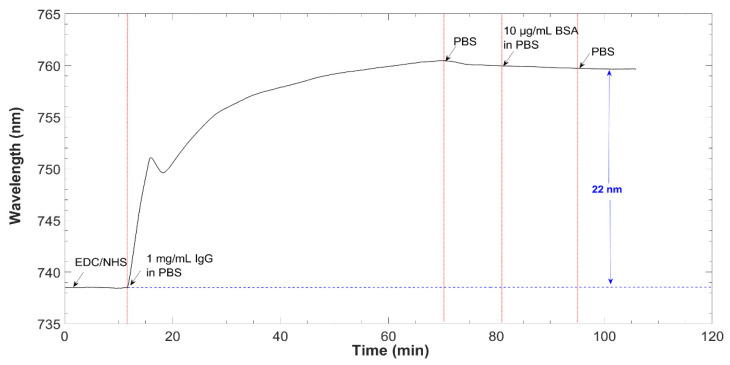
LMR central wavelength shift during IgG immobilization.

**Figure 10 biosensors-12-00403-f010:**
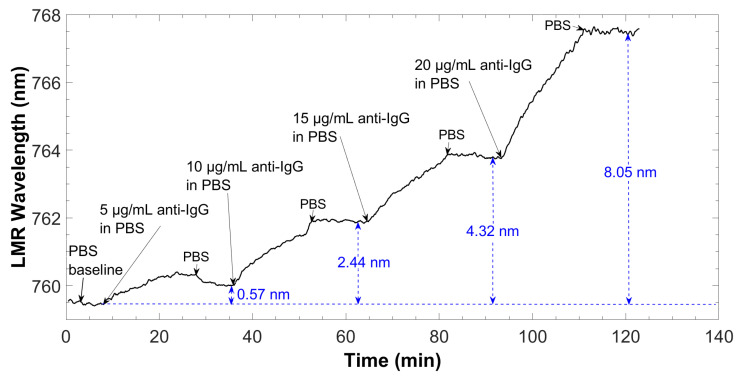
Sensorgram corresponding to the detection of different anti-IgG concentrations, starting from the baseline (blank in PBS buffer) up to 20 μg/mL of antigen solution.

**Figure 11 biosensors-12-00403-f011:**
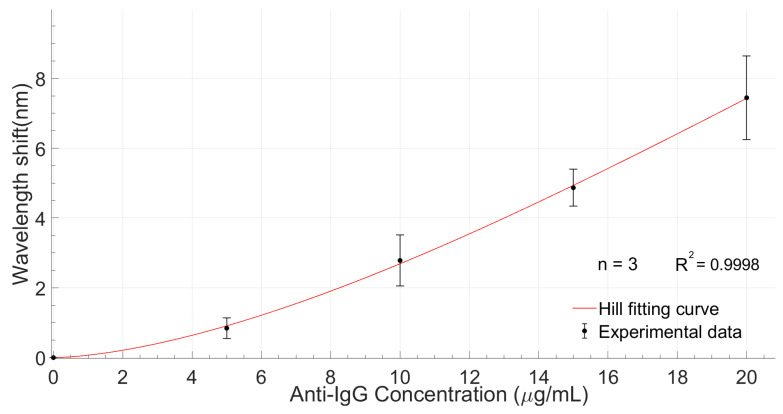
Calibration curve of three identical and independent biosensors (n = 3) detecting anti-IgG in PBS with the Hill fitting curve of the experimental points.

**Table 1 biosensors-12-00403-t001:** RI values and resonance wavelength shift corresponding to each glucose solution concentration.

Glucose Concentration (%)	RI	Wavelength Shift (nm)
10	1.346	22.60
20	1.358	44.94
30	1.370	67.92
40	1.379	88.14

## Data Availability

The data that support the findings of this study are available from the corresponding authors upon request.

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
