# Peer review of "Lossy Mode Resonance Based Microfluidic Platform Developed on Planar Waveguide for Biosensing Applications"

_biosensors, 2022, doi:10.3390/bios12060403_

Round 1

Reviewer 1 Report

The article reported by Melanys et al. entitled “LMR-based microfluidic platform developed on planar waveguide for biosensing applications” presents the experimental and theoretical investigation of microfluidic channel assisted planar waveguide for detection of anti-IgG. The LMR is achieved by TiO2 and improves the detection accuracy and sensitivity. All in all, this is a good piece of work supported by theoretical and experimental validation. The optimization of the physical parameters of the sensor leads to a good sensitivity of 0.43 nm/(μg/mL) along with LOD of 2.2 μg/mL, which turns the sensor into a promising candidate for anti-IgG detection.  The article is a good match to the goal of Biosensors, however, there are some  minor concerns that need to be explained:

1. The state of the art can be improved following latest review article: 10.3390/toxins13020089, 10.3390/bios12010042, 10.3390/bios12040195, etc.

2. Can the author discuss the reason for using TiO2 to achieve resonance instead of Au, Ag, etc. As Ag provides high detection accuracy with a sharp peak and Au is considered an ideal metal for biosensing.

3. Page 10, line 333, “which could be improved….” Authors are suggested to either provide the reference for the sentence or consider removing it from the article.

4. Authors are advised to discuss the measurement process, and how they avoid residue interaction during the monitoring, technical detail is a must.

5. Can the author explain how they align the MMFs. and how they fix the alignment during the experiment.

6.  How much measurand the microfluidic channel contains? Is the quantity fixed throughout the investigation? does the quantity affect the result?

7. Presentation and technical merit of the work seem quite satisfactory for the reader but grammar and syntax need to be improved.

Author Response

ANSWERS TO REVIEWER No.1

 Comment: The article reported by Melanys et al. entitled “LMR-based microfluidic platform developed on planar waveguide for biosensing applications” presents the experimental and theoretical investigation of microfluidic channel assisted planar waveguide for detection of anti-IgG. The LMR is achieved by TiO2 and improves the detection accuracy and sensitivity. All in all, this is a good piece of work supported by theoretical and experimental validation. The optimization of the physical parameters of the sensor leads to a good sensitivity of 0.43 nm/(μg/mL) along with LOD of 2.2 μg/mL, which turns the sensor into a promising candidate for anti-IgG detection. The article is a good match to the goal of Biosensors, however, there are some minor concerns that need to be explained:

Answer from the authors:

We would like to thank the Reviewer for the opportunity to enrich our paper with the proposed suggestions.

Comment 1: The state of the art can be improved following latest review article: 10.3390/toxins13020089, 10.3390/bios12010042, 10.3390/bios12040195, etc.

Answer from the authors:

We agree with the Reviewer that these articles can improve the information we provided. According to his/her suggestion we have included the references in the new version of the manuscript:

Page 1:

[2]       B. M. Azizur Rahman et al., “Optical Fiber, Nanomaterial, and THz-Metasurface-Mediated Nano-Biosensors: A Review,” Biosensors, vol. 12, no. 1, pp. 1–28, 2022, doi: 10.3390/bios12010042.

Page 2:

[15]     A. Nabok, A. M. Al-Jawdah, B. Gémes, E. Takács, and A. Székács, “An Optical Planar Waveguide-Based Immunosensors for Determination of Fusarium Mycotoxin Zearalenone,” Toxins (Basel)., vol. 13, no. 2, pp. 1–12, 2021, doi: 10.3390/toxins13020089.

[16]     P. A. Kocheril, K. D. Lenz, D. D. L. Mascareñas, J. E. Morales-Garcia, A. S. Anderson, and H. Mukundan, “Portable Waveguide-Based Optical Biosensor,” Biosensors, vol. 12, no. 4, p. 195, 2022, doi: 10.3390/bios12040195.

Comment 2: Can the author discuss the reason for using TiO2 to achieve resonance instead of Au, Ag, etc. As Ag provides high detection accuracy with a sharp peak and Au is considered an ideal metal for biosensing.

Answer from the authors:

We thank the Reviewer for letting us clarify this point. As he/she says, pure metals such as Au or Ag are excellent materials for developing biosensors. However, Lossy Mode Resonance phenomenon can be achieved by coating the surfaces with metal oxides and not pure metals. We are currently focused on LMR-based sensors since LMR complements the metallic materials typically used in surface plasmon resonance (SPR)–based sensors, with metal oxides and polymers. In addition, by depositing a thicker layer of these materials, it is possible to tune the position of the resonance in the optical spectrum or even to excite the resonance with both transverse electric (TE) and transverse magnetic (TM) polarized light, leading to the generation of multiple resonances. On the other hand, TiO2 has proved to have good biocompatibility and it has already been used for a broad range of medical applications.

We have added a new text in the revised manuscript so that this point can be better understood:

New text (from line 44 to 50)

According to this, materials which can induce this phenomenon are typically metal oxides and polymer coatings, which are less expensive than the metallic materials typically used in surface plasmon resonance (SPR)-based sensors. In addition, LMR enables one to tune the position of the resonance in the optical spectrum, to excite the resonance with both transverse electric (TE) and transverse magnetic (TM) polarized light, and to generate multiple resonances [7].

Comment 3:  Page 10, line 333, “which could be improved….” Authors are suggested to either provide the reference for the sentence or consider removing it from the article.

Answer from the authors:

According to the Reviewer’s comment, we have removed the cite expression from the article.

Comment 4: Authors are advised to discuss the measurement process, and how they avoid residue interaction during the monitoring, technical detail is a must.

Answer from the authors: We agree with the Reviewer in the sense that residue interaction can influence the measurement if they get to interact actually with the bioreceptor. However, both the proposed system and measurement protocol guarantee an adequate IgG – anti-IgG detection, as it is normally done in biosensing. In fact, Figure 10 can help understand these affirmations. First, once the IgGs have been immobilized, a phosphate buffer solution (PBS) is introduced in the microfluidic system to establish the baseline of the measurement. After that, the subsequent anti-IgGs detections are followed by PBS rinses. Since the detection times are always the same, PBS rinsing allows a double function: to remove all those anti-IgGs that have not bonded properly and to clean the available sensor surface and set the baseline for the next anti-IgGs concentration. If during this rinsing process, some air bubble, dust mote or even an undesired substance is introduced in the flow cell, the sensor will react with sudden changes due to refractive index variations. This can be avoided by adequately cleaning and sealing the microfluidics within the flow cell before proceeding with biosensing.

Comment 5: Can the author explain how they align the MMFs. and how they fix the alignment during the experiment.

Answer from the authors: We would like to thank the Reviewer the opportunity to address this point in more detail. The bottom part of the PMMA cell contains two slots for placing the MMFs and these slots are aligned with the sensor slot. One advantage of the proposed setup is that, due to the thickness of the MMFs and the coverslip, precise alignment systems are not needed.

Comment 6: How much measurand the microfluidic channel contains? Is the quantity fixed throughout the investigation? does the quantity affect the result?

Answer from the authors: We thank the Reviewer for letting us explain this issue in more detail. The microfluidic channel contains a volume of 10 µL. This value obviously depends on the channel dimensions, that are 10 mm x 1 mm x 1 mm. These values were chosen since they are the minimun required to produce the LMR phenomenon. According to this, the quantity is fixed throughout the investigation.

We have modified the text in the manuscript to make this point clear:

New text (from line 166 to 168)

The top part contains the holes for inlet and outlet fluids and a flow channel of 10 mm x 1 mm x 1 mm through which a liquid volume of 10 μL circulates.

Comment 7: Presentation and technical merit of the work seem quite satisfactory for the reader, but grammar and syntax need to be improved.

Answer from the authors: We appreciate the Reviewer’s comments and we have worked on improving the writing of the manuscript.

Reviewer 2 Report

See the file attached.

Author Response

ANSWERS TO REVIEWER No.2

Comment: In the present work, M. Benítez and co-authors well-described an optical biosensor based on planar waveguides on which they demonstrated it is possible to excite Lossy Mode Resonances (LMR). The authors provide a sufficiently clear description of the assembled setup and of the chemical procedure used to realize the integrated biosensor. The work is completed by an experimental calibration finalized to carry out the system sensitivity to sensing medium refractive index changes. Moreover, the authors demonstrated the potential application for biosensing of the system by showing an exemplary case of bioassay for detecting the IgG/anti-IgG binding.

Although the system shows, at the present stage, a low LoD, the authors reached the aim to demonstrate the potentiality of the technique. Apart from a few comments below, I consider the manuscript suitable for publication in this journal after minor revisions

Answer from the authors:

We appreciate the positive evaluation of the Reviewer.

Comment 1: Line 2 (Title). In my opinion, for clarity, the authors should avoid the acronym in the title.

Answer from the authors:

According with the Reviewer suggestion, we replaced the acronym in the title with the full name: Lossy Mode Resonance.

Comment 2: Line 82. Maybe the author should also cite the papers in the following:

  • Sinibaldi, A.; et al., Label-Free Monitoring of Human IgG/Anti-IgG Recognition Using Bloch Surface Waves on 1D Photonic Crystals. Biosensors 2018, 8, 71. https://doi.org/10.3390/bios8030071. Here, the authors used a label-free technique based on Bloch surface waves and a similar protocol for bioassay for detecting IgG/anti-IgG binding. They demonstrated a LoD in the order of 28 ng/mL for Anti-IgG antibody.
  • Q. Wang, Z. Wan-Ming, A comprehensive review of lossy mode resonance-based fiber optic sensors, Optics and Lasers in Engineering, 100(2018), 47-60, https://doi.org/10.1016/j.optlaseng.2017.07.009.

Answer from the authors:

We would like to thank the Reviewer for his/her suggestion. We have included the references in the new version of the manuscript:

Page 1:

[6]       Q. Wang and W. M. Zhao, “A comprehensive review of lossy mode resonance-based fiber optic sensors,” Opt. Lasers Eng., vol. 100, no. February 2017, pp. 47–60, 2018, doi: 10.1016/j.optlaseng.2017.07.009.

Page 5:

[28]     A. Sinibaldi, A. Occhicone, P. Munzert, N. Danz, F. Sonntag, and F. Michelotti, “Label-free monitoring of human IgG/anti-IgG recognition using Bloch surface waves on 1D photonic crystals,” Biosensors, vol. 8, no. 3, pp. 1–13, 2018, doi: 10.3390/bios8030071.

Comment 3: Line 120. In my opinion, the claim “The coverslip is placed into the microfluidic cell” is not clear here. It is clear after the microfluidic cell description.

Answer from the authors:

We thank the Reviewer for the focused observation. According to this, we have removed the first sentence of the paragraph and we have modified the second one to explain the use of Peltier cells and the thermistor. The details regarding the microfluidic cell are yet clearly described in section 2.4.

New text:

A temperature control system acquires the temperature from a thermistor and drives current to two Peltier cells in order to adjust the temperature at 26 °C (±0.05 °C) [20]. The microfluidic system is connected to a peristaltic pump that allows to pump the appropriate solution into the flow cell.

Comment 4: Line 139. Please, remove “a” before “the optimum design…”.

Answer from the authors:

Thank you for the correction. We have solved it accordingly.

Comment 5: Lines 245, 273. Please, correct the paragraph number sequence.

Answer from the authors: We appreciate the Reviewer observation. We have numbered the paragraphs correctly in this new version.

Comment 6: Line 350. Please correct “wavelenthg”.

Answer from the authors:                                  

We would like to thank the Reviewer for this observation. We have corrected the mistake. 

ANSWERS TO REVIEWER No.2

Comment: In the present work, M. Benítez and co-authors well-described an optical biosensor based on planar waveguides on which they demonstrated it is possible to excite Lossy Mode Resonances (LMR). The authors provide a sufficiently clear description of the assembled setup and of the chemical procedure used to realize the integrated biosensor. The work is completed by an experimental calibration finalized to carry out the system sensitivity to sensing medium refractive index changes. Moreover, the authors demonstrated the potential application for biosensing of the system by showing an exemplary case of bioassay for detecting the IgG/anti-IgG binding.

Although the system shows, at the present stage, a low LoD, the authors reached the aim to demonstrate the potentiality of the technique. Apart from a few comments below, I consider the manuscript suitable for publication in this journal after minor revisions

Answer from the authors:

We appreciate the positive evaluation of the Reviewer.

Comment 1: Line 2 (Title). In my opinion, for clarity, the authors should avoid the acronym in the title.

Answer from the authors:

According with the Reviewer suggestion, we replaced the acronym in the title with the full name: Lossy Mode Resonance.

Comment 2: Line 82. Maybe the author should also cite the papers in the following:

  • Sinibaldi, A.; et al., Label-Free Monitoring of Human IgG/Anti-IgG Recognition Using Bloch Surface Waves on 1D Photonic Crystals. Biosensors 2018, 8, 71. https://doi.org/10.3390/bios8030071. Here, the authors used a label-free technique based on Bloch surface waves and a similar protocol for bioassay for detecting IgG/anti-IgG binding. They demonstrated a LoD in the order of 28 ng/mL for Anti-IgG antibody.
  • Q. Wang, Z. Wan-Ming, A comprehensive review of lossy mode resonance-based fiber optic sensors, Optics and Lasers in Engineering, 100(2018), 47-60, https://doi.org/10.1016/j.optlaseng.2017.07.009.

Answer from the authors:

We would like to thank the Reviewer for his/her suggestion. We have included the references in the new version of the manuscript:

Page 1:

[6]       Q. Wang and W. M. Zhao, “A comprehensive review of lossy mode resonance-based fiber optic sensors,” Opt. Lasers Eng., vol. 100, no. February 2017, pp. 47–60, 2018, doi: 10.1016/j.optlaseng.2017.07.009.

Page 5:

[28]     A. Sinibaldi, A. Occhicone, P. Munzert, N. Danz, F. Sonntag, and F. Michelotti, “Label-free monitoring of human IgG/anti-IgG recognition using Bloch surface waves on 1D photonic crystals,” Biosensors, vol. 8, no. 3, pp. 1–13, 2018, doi: 10.3390/bios8030071.

Comment 3: Line 120. In my opinion, the claim “The coverslip is placed into the microfluidic cell” is not clear here. It is clear after the microfluidic cell description.

Answer from the authors:

We thank the Reviewer for the focused observation. According to this, we have removed the first sentence of the paragraph and we have modified the second one to explain the use of Peltier cells and the thermistor. The details regarding the microfluidic cell are yet clearly described in section 2.4.

New text:

A temperature control system acquires the temperature from a thermistor and drives current to two Peltier cells in order to adjust the temperature at 26 °C (±0.05 °C) [20]. The microfluidic system is connected to a peristaltic pump that allows to pump the appropriate solution into the flow cell.

Comment 4: Line 139. Please, remove “a” before “the optimum design…”.

Answer from the authors:

Thank you for the correction. We have solved it accordingly.

Comment 5: Lines 245, 273. Please, correct the paragraph number sequence.

Answer from the authors: We appreciate the Reviewer observation. We have numbered the paragraphs correctly in this new version.

Comment 6: Line 350. Please correct “wavelenthg”.

Answer from the authors:                                  

We would like to thank the Reviewer for this observation. We have corrected the mistake.